# Deep template-based protein structure prediction

**Fandi Wu**[1,2,3], **Jinbo Xu**[1]*

**1** Toyota Technological Institute at Chicago, Chicago, IL, United States of America, **2** Institute of Computing Technology, Chinese Academy of Sciences, Beijing, China, **3** University of Chinese Academy of Sciences, Beijing, China

* jinboxu@gmail.com

## Abstract

### Motivation

Protein structure prediction has been greatly improved by deep learning, but most efforts are devoted to template-free modeling. But very few deep learning methods are developed for TBM (template-based modeling), a popular technique for protein structure prediction. TBM has been studied extensively in the past, but its accuracy is not satisfactory when highly similar templates are not available.

### Results

This paper presents a new method NDThreader (New Deep-learning Threader) to address the challenges of TBM. NDThreader first employs DRNF (deep convolutional residual neural fields), which is an integration of deep ResNet (convolutional residue neural networks) and CRF (conditional random fields), to align a query protein to templates without using any distance information. Then NDThreader uses ADMM (alternating direction method of multipliers) and DRNF to further improve sequence-template alignments by making use of predicted distance potential. Finally, NDThreader builds 3D models from a sequence-template alignment by feeding it and sequence coevolution information into a deep ResNet to predict inter-atom distance distribution, which is then fed into PyRosetta for 3D model construction. Our experimental results show that NDThreader greatly outperforms existing methods such as CNFpred, HHpred, DeepThreader and CEthreader. NDThreader was blindly tested in CASP14 as a part of RaptorX server, which obtained the best average GDT score among all CASP14 servers on the 58 TBM targets.

**Data Availability Statement:** The training, validation and test protein lists are available in S4 File. The code is available for download at http://raptorx.uchicago.edu/download/ and https://github.com/wufandi/DL4SequenceAlignment.

## Author summary

TBM (template-based modeling) is a popular method for protein structure prediction. However, existing methods cannot generate good models when the protein under prediction does not have very similar templates in Protein Data Bank (PDB). Recently significant progress has been made on template-free protein structure prediction by deep learning,

**Funding:** This work is supported by National Institutes of Health grant R01GM089753 to J.X. and National Science Foundation grant DBI1564955 to J.X. This work is also supported by the CSC Scholarship, the National Key Research and Development Program of China (under Grant 2020AAA0103802), and the NSF of China (U20A20227) to F.W. The funders had no role in study design, data collection and analysis, decision to publish, or preparation of the manuscript.

but very few deep learning methods were developed for TBM. To further improve TBM for protein structure prediction, we present a new deep learning method that greatly outperforms existing ones in identifying the best templates, generating sequence-template alignment and constructing 3D models from alignments. Blindly tested in CASP14, our server obtained the best average model quality score on the 58 TBM targets among all the CASP14-participating servers, which confirms that our method is effective for TBM.

## Introduction

Predicting protein structure from its amino acid sequence is one of the most challenging problems in the field of computational biology. Template-based modeling (TBM), including protein threading and homology modeling, is a popular method for protein tertiary structure prediction. TBM predicts the structure of a query protein (called target) by aligning it to one or multiple templates with solved structures. Along with the growth of the PDB (Protein Data Bank), TBM is able to predict structures for a good percentage of proteins [1]. For example, in CASP13 67 out of 112 test domains and in CASP14 58 out of 107 test domains have reasonable templates in PDB. When a protein under prediction does not have highly similar templates, TBM faces three major challenges: selection of the best templates, building an accurate sequence-template alignment, and constructing 3D models from the alignment.

TBM uses a scoring function to guide sequence-template alignment. Existing methods such as HHpred [2] and SPARKS-X [3] employ a linear scoring function composed of sequential features such as sequence profile, predicted secondary structure and solvent accessibility. CNFpred [4] uses a CRF (conditional random fields) plus a shallow CNN (convolutional neural network) to learn a scoring function from reference alignments. Due to relatively simple scoring functions, these methods cannot capture the complex relationship between input features and protein alignment. As a result of recent progress on contact/distance prediction [5–7], predicted contact and distance have been explored to improve protein alignments. For example, map_align [8], EigenThreader [9] and CEthreader [10] make use of predicted contacts to improve alignments. DeepThreader [10] is the first TBM method that makes use of predicted inter-residue distance potential. Different from the well-known protein structure alignment program DALI [11] that can only take distance matrices as input, DeepThreader may take both distance probability distribution (of the query protein) and distance matrix as input. In addition, DeepThreader takes other input features (e.g., predicted secondary structure and sequence profile) as input, but DALI does not. Blindly tested in CASP13 as a server RaptorX-TBM [5], DeepThreader outperformed all the other pure threading-based servers and on the very hard targets even performed comparably to Robetta that utilized a combination of TBM, fragment-based and contact-assisted folding. Nevertheless, DeepThreader's performance is still not very satisfactory because 1) it uses a shallow convolutional neural network method CNFpred to generate an initial alignment, which cannot be improved much by DeepThreader when it is of low quality; 2) the predicted distance potential used by DeepThreader in CASP13 is not accurate enough.

Deep learning has greatly improved template-free protein modeling, but not much effort has been devoted to study deep learning methods for template-based modeling. Since our proposal of deep ResNet for protein contact/distance prediction and template-free protein modeling [5–7,11–13], deep ResNet has been widely used by many groups to predict inter-residue (-atom) relationships, such as DMPfold [14], AlphaFold1 [15] and trRosetta [16]. Here we show that deep ResNet works well for protein alignments even in the absence of predicted

contact/distance information. In particular, we integrate deep ResNet and CRF (Conditional Random Fields) [17] to form a new deep network DRNF (Deep Convolutional Residual Neural Fields) that may capture context-specific information from sequential features to improve protein alignment. Our experimental results show that DRNF generates better alignments and recognizes better templates than existing methods such as HHpred and CNFpred. When DRNF is combined with predicted distance potential, it may further improve alignment accuracy over an existing distance-based threading method DeepThreader, as evidenced by our results on the CASP13 and CAMEO data.

MODELLER and RosettaCM are often used to build 3D models from a sequence-template alignment, but they usually generate 3D models more similar to the templates than to the protein under prediction. To address this issue, we build 3D models from an alignment using our own method by combining template and sequence coevolution information. In particular, we feed a sequence-template alignment and sequence coevolution information into a deep ResNet to predict inter-residue distance/orientation distribution, convert the distribution to distance/orientation potential and then minimize the potential through PyRosetta [18] to build 3D models. In CASP14 our method obtained the best average GDT score among all CASP14-participating servers on the 58 TBM targets.

## Results

### Overview of the method

Fig 1 shows the overall architecture of our method, which mainly consists of three modules. The first is a DRNF (Deep Convolutional Residual Neural Fields) module for query-template alignment without using any distance information. DRNF uses deep ResNet to capture context-specific information from sequential features and integrate it with CRF (Conditional Random Fields) to predict query-template alignments. The second module employs ADMM [13,19] and another deep ResNet to improve the DRNF-generated alignments by making use of predicted distance potential of the query. Finally, the sequence-template alignment, the template and the sequence coevolution information are fed into a deep ResNet to predict inter-atom distance/orientation probability distribution, which is then fed into PyRosetta to build 3D models [20].

Here we use DRNF and NDThreader (New Deep-learning Threader) to denote our alignment methods without and with predicted distance information, respectively. To evaluate alignment accuracy and threading performance, we compare our methods with several established methods including HHpred, CNFpred, CEThreader and DeepThreader. HHpred was run with two alignment modes: global with option '-mact 0' and local with option '-mact 0.1'. Since DeepThreader is the first distance-based threading method and also performed the best among all pure threading-based methods in CASP13, we pay more attention to the comparison with DeepThreader.

### Evaluation of alignment quality

**Alignment accuracy on in-house test set when distance information is not used.** We train DRNF on DeepAlign-generated alignments (see Methods for details) and use two methods Viterbi and MaxAcc (maximum expected accuracy) to predict query-template alignments in the absence of distance information. We divide the test set into 4 bins according to the query-template structure similarity (TMscore): (0.45, 0.55], (0.55, 0.65], (0.65, 0.8] and (0.8, 1]. Table 1 shows that DRNF outperforms CNFpred and HHpred by a good margin in terms of both reference-dependent recall and precision especially when the sequence-template structure similarity is not very high. Even if trained by DeepAlign-generated alignments, DRNF

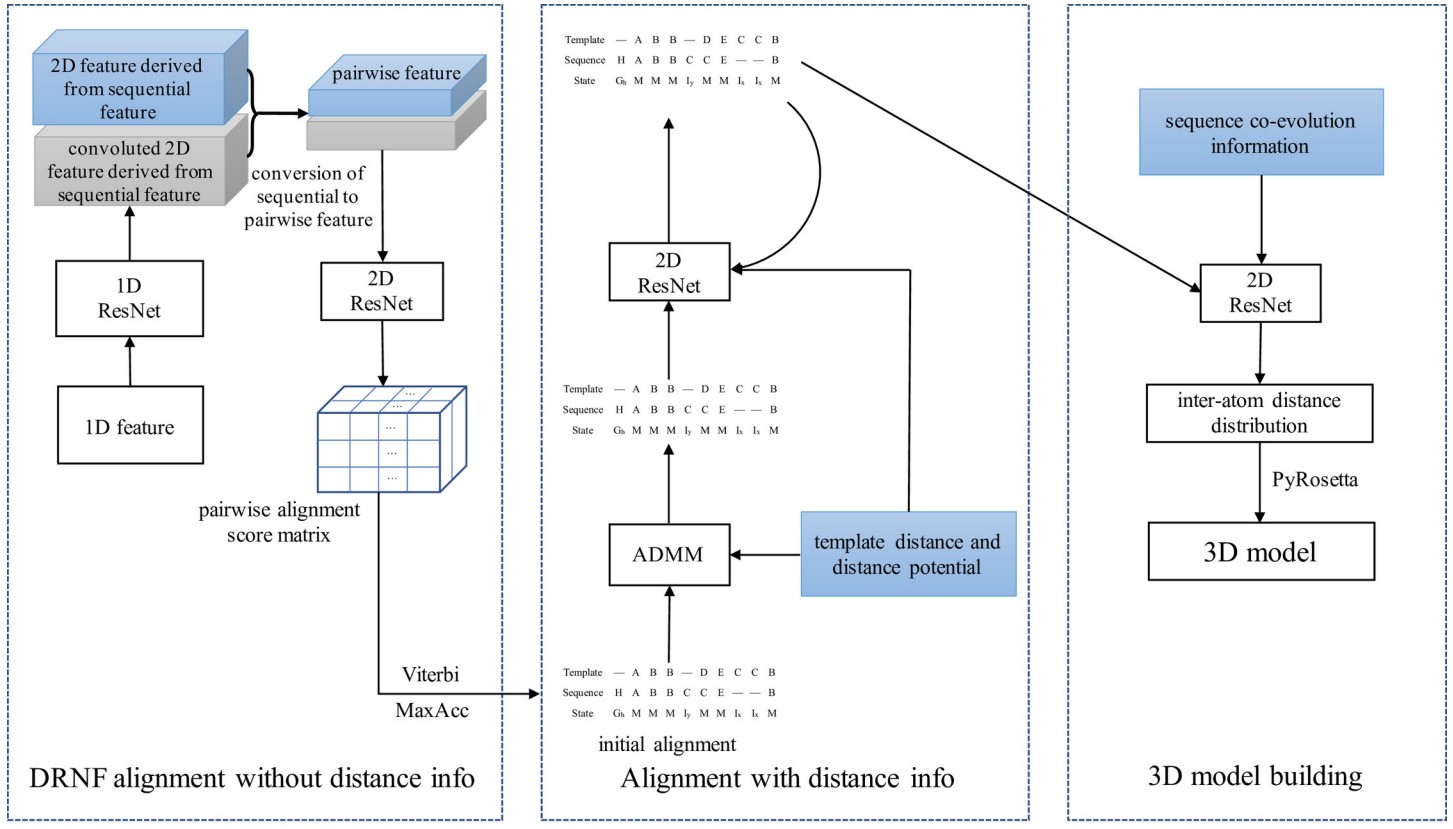

**Fig 1. The overall architecture of our template-based modelling method.**

still has the best precision and recall when evaluated by the TMalign-generated reference alignments. Table 2 and Fig 2 show that DRNF also outperforms CNFpred and HHpred in terms of reference-independent accuracy of the 3D models built by MODELLER [21] from sequence-template alignments. The DRNF-generated alignments have average TMscore and GDT 0.525

**Table 1. Reference-dependent alignment accuracy (precision and recall) on our in-house benchmark.**

| | CNFpred | | HHpred-global | | HHpred-local | | DRNF-Viterbi | | DRNF-MaxAcc | |
|---|---|---|---|---|---|---|---|---|---|---|
| | recall | prec | recall | prec | recall | prec | recall | prec | recall | prec |
| | Evaluated by DeepAlign-generated reference alignments | | | | | | | | | |
| (0,1] | 0.475 | 0.474 | 0.395 | 0.344 | 0.351 | 0.475 | 0.616 | 0.615 | 0.596 | 0.635 |
| (0.45,0.55] | 0.240 | 0.236 | 0.141 | 0.111 | 0.118 | 0.268 | 0.382 | 0.377 | 0.353 | 0.394 |
| (0.55,0.65] | 0.422 | 0.426 | 0.338 | 0.282 | 0.284 | 0.435 | 0.591 | 0.594 | 0.579 | 0.628 |
| (0.65,0.8] | 0.601 | 0.596 | 0.530 | 0.468 | 0.470 | 0.580 | 0.748 | 0.747 | 0.732 | 0.763 |
| (0.8,1] | 0.830 | 0.829 | 0.778 | 0.732 | 0.774 | 0.865 | 0.848 | 0.845 | 0.816 | 0.837 |
| | Evaluated by TMalign-generated reference alignments | | | | | | | | | |
| (0,1] | 0.436 | 0.426 | 0.318 | 0.358 | 0.435 | 0.322 | 0.545 | 0.532 | 0.526 | 0.548 |
| (0.45,0.55] | 0.206 | 0.194 | 0.094 | 0.115 | 0.221 | 0.094 | 0.313 | 0.292 | 0.288 | 0.308 |
| (0.55,0.65] | 0.377 | 0.369 | 0.255 | 0.296 | 0.391 | 0.252 | 0.510 | 0.497 | 0.495 | 0.521 |
| (0.65,0.8] | 0.567 | 0.555 | 0.436 | 0.487 | 0.545 | 0.436 | 0.676 | 0.666 | 0.656 | 0.676 |
| (0.8,1] | 0.779 | 0.780 | 0.704 | 0.751 | 0.756 | 0.865 | 0.815 | 0.814 | 0.801 | 0.825 |

**Table 2. Reference-independent alignment quality measured by TM-score and GDT on our in-house test set.** GDT is scaled to [0, 1]. DRNF is trained by DeepAlign-generated alignments and uses Viterbi to build alignments.

| | CNFpred | | HHpred-global | | HHpred-local | | DRNF | |
|---|---|---|---|---|---|---|---|---|
| | TMscore | GDT | TMscore | GDT | TMscore | GDT | TMscore | GDT |
| (0,1] | 0.469 | 0.383 | 0.415 | 0.338 | 0.341 | 0.290 | 0.525 | 0.432 |
| (0.45,0.55] | 0.320 | 0.244 | 0.232 | 0.176 | 0.156 | 0.131 | 0.380 | 0.294 |
| (0.55,0.65] | 0.426 | 0.331 | 0.374 | 0.290 | 0.283 | 0.234 | 0.493 | 0.389 |
| (0.65,0.8] | 0.554 | 0.465 | 0.512 | 0.427 | 0.442 | 0.377 | 0.610 | 0.515 |
| (0.8,1] | 0.709 | 0.635 | 0.691 | 0.614 | 0.673 | 0.596 | 0.723 | 0.648 |

and 0.432, respectively, outperforming HHpred and CNFpred by a large margin. DRNF generates better alignments than HHpred and CNFpred for 852 and 781 out of 1000 protein pairs, respectively. The t-test shows that the difference between DRNF and HHpred is statistically highly significant (P-value = 8.2E-36), so is the difference between DRNF and CNFpred (P-value = 8.5E-15).

By the way, we have trained several different DRNF models under different settings (e.g., different input features and reference alignments) and analyzed their performance. See sections F and G in S1 File for the training curves and ablation study. Section H in S1 File also shows the relationship between DRNF alignment score and reference-independent alignment quality.

**Alignment accuracy on the CASP13 data when distance information is used.** Fig 3 summarizes the average reference-independent alignment accuracy (TMscore and GDT) of our methods DRNF and NDThreader and shows their comparison with DeepThreader and CNFpred. Here all the 3D models are built by MODELLER from the sequence-template alignments generated by these competing methods. On average, the alignments produced by NDThreader have TMscore and GDT of 0.624 and 0.556. In terms of TMscore, NDThreader outperforms DeepThreader, DRNF and CNFpred by 6.1%, 12.4% and 19.9%, respectively. NDThreader outperforms DeepThreader on ~490 protein pairs, whereas DeepThreader outperforms NDThreader on only ~270 pairs. For those protein pairs with TMscore<0.6, NDThreader has a much larger advantage over DeepThreader. NDThreader generates better alignment than DRNF for ~650 (out of 764) pairs, which confirms that the predicted distance potential is very useful. The difference between DRNF and CNFpred is statistically significant

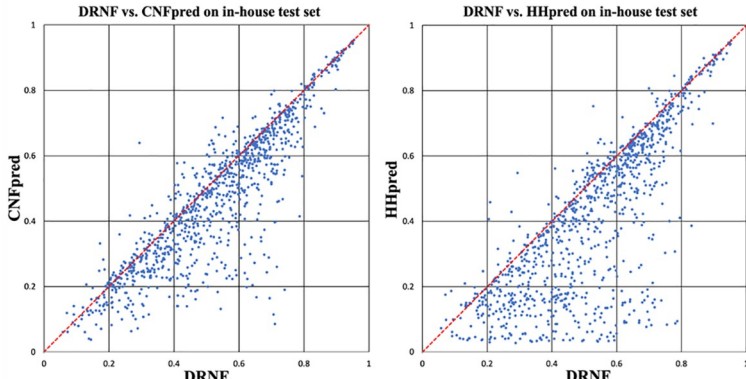

**Fig 2. The TMscore of the 3D models built from the alignments generated by DRNF, CNFpred and HHpred on our in-house test set.** (left) DRNF vs. CNFpred; (right) DRNF vs. HHpred. Each point represents two alignments generated by two competing methods for the same protein pair.

|  | NDThreader | DeepThreader | DRNF | CNFpred |
|---|---|---|---|---|
| TM-score | 0.557 | 0.53 | 0.515 | 0.484 |
| GDT | 0.459 | 0.436 | 0.421 | 0.395 |

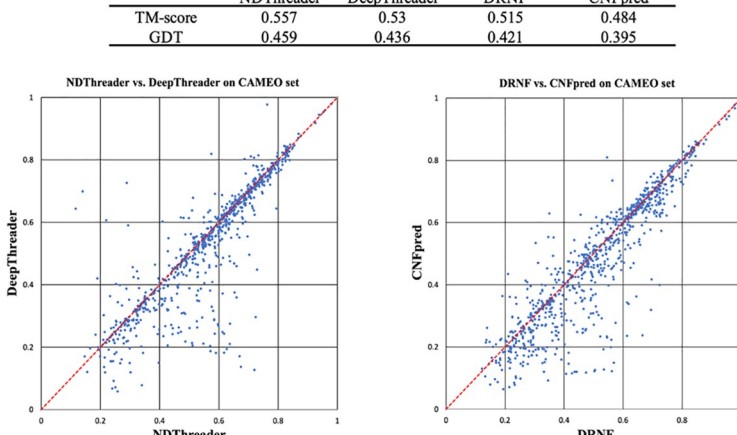

**Fig 3. Alignment quality (TM-score and GDT) comparison between DRNF, CNFpred, NDThreader and DeepThreader on the CASP13 alignment test set.** Top: average alignment quality (TM-score and GDT) on the CASP13 alignment test set. GDT is scaled to [0, 1]. Bottom left: NDThreader vs. DeepThreader. Bottom right: DRNF vs. CNFpred. Each point represents the quality of two alignments generated by two competing methods for the same protein pair.

(P-value = 0.005), so is the difference between NDThreader and DeepThreader (P-value = 0.001).

**Alignment accuracy on the CAMEO data when distance information is used.** Here we evaluate our methods DRNF and NDThreader in terms of reference-independent quality (TMscore and GDT) on the CAMEO test set. Again, all the 3D models are built by MODEL-LER from alignments. Fig 4 lists the average alignment quality and shows their head-to-head comparison with CNFpred and DeepThreader. On average NDThreader has the best alignment quality, 0.027 better than DeepThreader in terms of TMscore. DRNF has an average TM-score 0.515, 0.031 higher than CNFpred. The average score of NDThreader is not much higher than DeepThreader because many test pairs have very similar proteins and any

|  | NDThreader | DeepThreader | DRNF | CNFpred | HHpred |
|---|---|---|---|---|---|
| TM-score | 0.624 | 0.59 | 0.557 | 0.522 | 0.468 |
| GDT | 0.556 | 0.525 | 0.495 | 0.466 | 0.416 |

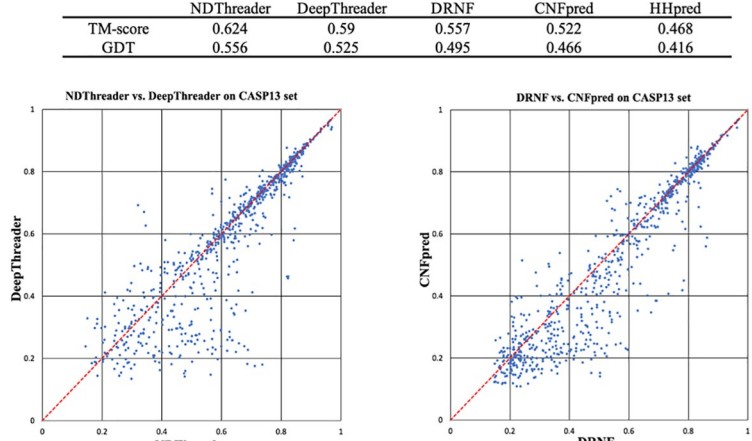

**Fig 4. Alignment quality (TM-score and GDT) comparison between DRNF, CNFpred, NDThreader and DeepThreader on the CAMEO test set.** Top: average alignment quality (TM-score and GDT) on the CAMEO test set. GDT is scaled to [0, 1]. Bottom left: NDThreader vs. DeepThreader. Bottom right: DRNF vs. CNFpred. Each point represents two alignments generated by two competing methods for the same protein pair.

methods can do well on them. Fig 4 shows that NDThreader generates alignment better than DeepThreader on ~570 protein pairs, whereas DeepThreader does better on ~230 pairs. NDThreader generates better alignment than DRNF on ~630 protein pairs, whereas DRNF is better than NDThreader on only ~190 pairs. The difference between DRNF and CNFpred is statistically significant with P-value = 0.002, so is the difference between NDThreader and DeepThreader (P-value = 0.001).

## Evaluation of threading performance

**Threading performance on the CASP13 targets.** We evaluate the threading performance of our methods in Table 3. See S2 File for detailed accuracy. Here all the 3D models are built by MODELLER from the alignments and only a single template is used to build one 3D model. CASP13 has 45 TBM-easy, 22 TBM-hard, 32 FM and 13 FM/TBM domains, respectively. In this benchmark NDThreader and DeepThreader use the same set of distance information predicted by our deep ResNet method for the query proteins. The same template database is used for NDThreader, DRNF, DeepTheader and CNFpred.

On the FM targets, NDThreader outperforms DeepThreader by 32.4% and 31.5%, respectively, in terms of TMscore and GDT, when the first-ranked models are evaluated. When the best of the top 5 templates are evaluated, NDThreader is 30% and 28.6% better than Deep-Threader in terms of TM-score and GDT, respectively. Fig 5 shows their head-to-head comparison when the first-ranked and the best of top 5 templates are considered. The best of top 5 models produced by NDThreader for the FM targets have an average TM-score 0.473, 8.2% higher than the first-ranked models. In terms of TM-score, NDThreader ranks the best of top 5 models first for only 9 out of 32 FM domains, which indicates that template selection by the raw alignment score is not very accurate for the FM targets.

On the FM/TBM targets, DRNF is better than CNFpred by 0.018 TMscore and NDThreader is better than DeepThreader by 0.10 TMscore when the first-ranked models are evaluated.

**Table 3. The threading performance on the CASP13 targets.** GDT is scaled to [0, 1].

| | top 1 | | | best of top 5 | | | best of top 50 | | |
|---|---|---|---|---|---|---|---|---|---|
| | TM | GDT | (TM+GDT)/2 | TM | GDT | (TM+GDT)/2 | TM | GDT | (TM+GDT)/2 |
| | | | | 45 TBM-Easy targets | | | | | |
| NDThreader | 0.819 | 0.764 | 0.792 | 0.826 | 0.773 | 0.800 | 0.827 | 0.772 | 0.800 |
| DeepThreader | 0.816 | 0.762 | 0.789 | 0.824 | 0.770 | 0.797 | 0.826 | 0.774 | 0.800 |
| DRNF | 0.791 | 0.736 | 0.764 | 0.807 | 0.754 | 0.781 | 0.816 | 0.765 | 0.790 |
| CNFpred | 0.785 | 0.732 | 0.758 | 0.802 | 0.750 | 0.776 | 0.805 | 0.753 | 0.779 |
| | | | | 22 TBM-Hard targets | | | | | |
| NDThreader | 0.716 | 0.612 | 0.664 | 0.743 | 0.630 | 0.687 | 0.748 | 0.635 | 0.692 |
| DeepThreader | 0.654 | 0.547 | 0.600 | 0.678 | 0.568 | 0.623 | 0.701 | 0.589 | 0.645 |
| DRNF | 0.619 | 0.510 | 0.564 | 0.655 | 0.543 | 0.599 | 0.672 | 0.563 | 0.617 |
| CNFpred | 0.599 | 0.499 | 0.549 | 0.650 | 0.542 | 0.596 | 0.657 | 0.554 | 0.605 |
| | | | | 13 FM/TBM targets | | | | | |
| NDThreader | 0.578 | 0.550 | 0.564 | 0.604 | 0.581 | 0.593 | 0.617 | 0.588 | 0.603 |
| DeepThreader | 0.478 | 0.449 | 0.463 | 0.546 | 0.510 | 0.528 | 0.581 | 0.549 | 0.565 |
| DRNF | 0.429 | 0.408 | 0.418 | 0.462 | 0.437 | 0.449 | 0.507 | 0.478 | 0.492 |
| CNFpred | 0.411 | 0.390 | 0.400 | 0.445 | 0.422 | 0.433 | 0.484 | 0.469 | 0.476 |
| | | | | 32 FM targets | | | | | |
| NDThreader | 0.437 | 0.380 | 0.408 | 0.473 | 0.405 | 0.439 | 0.491 | 0.423 | 0.457 |
| DeepThreader | 0.330 | 0.289 | 0.309 | 0.369 | 0.322 | 0.345 | 0.410 | 0.353 | 0.381 |

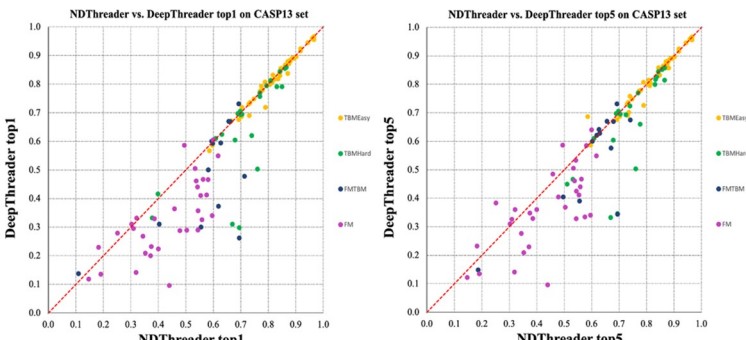

**Fig 5. Head-to-head comparison between NDThreader and DeepThreader on the CASP13 targets.** Left: top 1 models. Right: the best of top 5 models. Each point represents the quality (TM-score) of two models generated by NDThreader (x-axis) and DeepThreader (y-axis), respectively.

DeepThreader fails to produce good alignments for T0986s1, and to select good templates for T1008 and T0970. In terms of TMscore, NDThreader outperforms DRNF by 30% and Deep-Threader outperforms CNFpred by 16%, which confirms that predicted distance indeed can greatly improve threading.

On the TBM-hard targets, NDThreader is better than DeepThreader by 0.055 TMscore when the first-ranked models are evaluated, and NDThreader only underperforms Deep-Threader on T0979-D1 due to incorrect template selection. But when the best of the top 5 templates is considered, NDThreader has a good model for T0979-D1. DRNF outperforms CNFpred by 0.02 TMscore, and predicts better first-ranked models than CNFpred for 14 of 22 TBM-hard targets.

On the TBM-easy targets, NDThreader, DeepThreader, DRNF and CNFpred produce the first-ranked models with average TMscore 0.819, 0.816, 0.791 and 0.785, respectively. That is, even on easy targets predicted distance potential still helps slightly. When the first-ranked models are evaluated, DRNF outperforms CNFpred on 26 domains, while underperforms on 19 domains., NDThreader outperforms DeepThreader by 0.003 and 0.002 in terms of TMscore and GDT, respectively. In terms of the TMscore, NDThreader outperforms DeepThreader and DRNF on 26 and 36 domains, respectively.

In summary, NDThreader generates better first-ranked models than DeepThreader on 76 of 112 domains and better best-of-top-5 models on 79 domains. On average NDThreader has the best threading performance on all CASP13 targets. The t-test shows that the difference between NDThreader and DeepThreader statistically is not very significant (P-value = 0.066 and 0.073 when the first-ranked and the best of top 5 models are considered, respectively), possibly because the number of test protein domains is not very large. We also evaluate the NDThreader result against structurally the most similar templates detected by TMalign, as shown in section I in S1 File. This result indicates that for TBM targets, the NDThreader result is not far away from TMalign, but for FM and FM/TBM targets, there is still a gap.

**Comparison with top CASP13 servers.** Table 4 summarizes the performance of our new methods NDThreader and DRNF and some top CASP13 servers on the CASP13 targets. RaptorX-TBM [5] and CETheader are two servers tested in CASP13 and mainly based upon pure threading methods. RaptorX-TBM used DeepThreader to select templates and generate alignments and then used RosettaCM [22] to build 3D models. RaptorX-TBM used PDB90 as the template database while both DRNF and NDThreader use PDB40 created before CASP13. For some targets RaptorX-TBM used multiple templates to build 3D models, but NDThreader only uses a single template. CETheader is a contact-assisted threading method, but it is

**Table 4. Threading performance on 32 CASP13 FM, 13 FM/TBM, 22 TBM-Hard and 45 TBM-Easy domains.**

| | FM | | FM/TBM | | TBM-Hard | | TBMEasy | |
|---|---|---|---|---|---|---|---|---|
| | TMscore | GDT | TMscore | GDT | TMscore | GDT | TMscore | GDT |
| NDThreader (this work) | 0.44/0.47 | 37.98/40.51 | 0.58/0.60 | 55.02/58.08 | 0.72/0.74 | 61.17/63.01 | 0.82/0.83 | 76.40/77.24 |
| DRNF (this work) | 0.26/0.32 | 22.51/26.98 | 0.43/0.46 | 40.77/43.7 | 0.61/0.65 | 50.79/54.33 | 0.79/0.81 | 73.59/75.40 |
| CEThreader | 0.33/0.37 | 27.84/31.45 | 0.51/0.53 | 49.74/54.53 | 0.60/0.63 | 49.87/53.14 | 0.74/0.78 | 67.75/72.46 |
| RaptorX-TBM | 0.41/0.42 | 35.12/36.45 | 0.55/0.56 | 53.28/54.3 | 0.69/0.71 | 58.96/60.71 | 0.82/0.82 | 77.01/77.22 |
| RaptorX-DeepModeller | 0.47/0.5 | 41.24/43.86 | 0.58/0.60 | 56.49/58.57 | 0.68/0.69 | 58.99/59.92 | 0.83/0.84 | 78.12/79.18 |
| Zhang-Server | 0.49/0.52 | 42.78/46.03 | 0.60/0.64 | 57.68/61.5 | 0.72/0.75 | 62.25/64.51 | 0.83/0.85 | 78.23/79.82 |
| QUARK | 0.49/0.52 | 43.36/45.13 | 0.59/0.66 | 58.03/63.03 | 0.71/0.75 | 60.96/64.64 | 0.83/0.85 | 78.13/79.79 |

unclear how its 3D models were built. NDThreader and DRNF build 3D models from alignments using MODELLER, which is slightly worse than RosettaCM. NDThreader outperforms RaptorX-TBM and CEThreader on the FM, FM/TBM and TBM-Hard targets. On the TBM-Easy targets, NDThreader has a similar performance as RaptorX-TBM. On FM, FM/TBM and TBM-Hard targets DRNF is not comparable to RaptorX-TBM and NDThreader because DRNF does not use any distance information. RaptorX-DeepModeller [5], Zhang-Server and QUARK [23,24] used a mix of template-based (one and multiple templates) and template-free techniques to build 3D models. They outperformed NDThreader on the FM targets, but did not show significant advantage on the TBM targets.

## Performance in CASP14

We blindly tested our methods (as a part of RaptorX server) in CASP14, in which we employed NDThreader and DRNF to find the best templates for a TBM target (judged by HHpred E-value<1E-5) and built the sequence-template alignments. Instead of building 3D models using MODELLER and RosettaCM, we built 3D models using our own folding engine originally designed for template-free modeling [20]. In particular, we fed the coevolution information of the test target, its alignment with the selected template and the template distance matrix into our deep ResNet to predict the inter-atom distance and orientation probability distribution. Then we converted the predicted distribution into distance/orientation potential and used the gradient descent method in PyRosetta to build 3D models by minimizing the predicted potential. In fact, we initiated this idea in CASP13 [5] and further improved its implementation in CASP14. See S3 File for detailed modeling accuracy of our method, Zhang-Server and Baker-RosettaServer. Table 5 shows that in terms of TMscore our server performed similarly as the other two top servers and in terms of GDT our server did slightly better. Fig 6 shows the head-to-head comparison between RaptorX and Zhang-Server and between RaptorX and Rosetta-Server. In terms of TMscore, RaptorX outperforms Zhang-Server and RosettaServer on 28 and 24 targets (out of 58), respectively, and underperforms on 22 and 27 targets, respectively. In terms of GDT score, RaptorX outperforms Zhang-Server and BAKER-RosettaServer on 36 and 28 targets, respectively, while underperforms on 22 and 30 targets, respectively. RaptorX

**Table 5. The performance of three top servers on the CASP14 TBM targets.** Each entry has the average quality score of the 1st-ranked and the best of top 5 models.

| | 27 TBM-Easy targets | | 31 TBM-Hard targets | |
|---|---|---|---|---|
| | TMscore | GDT | TMscore | GDT |
| RaptorX | 0.860/0.864 | 79.44/80.30 | 0.697/0.732 | 62.23/65.35 |
| Zhang-Server | 0.854/0.859 | 77.92/78.50 | 0.696/0.722 | 61.69/64.07 |
| RosettaServer | 0.837/0.846 | 77.32/78.30 | 0.716/0.725 | 63.49/64.75 |

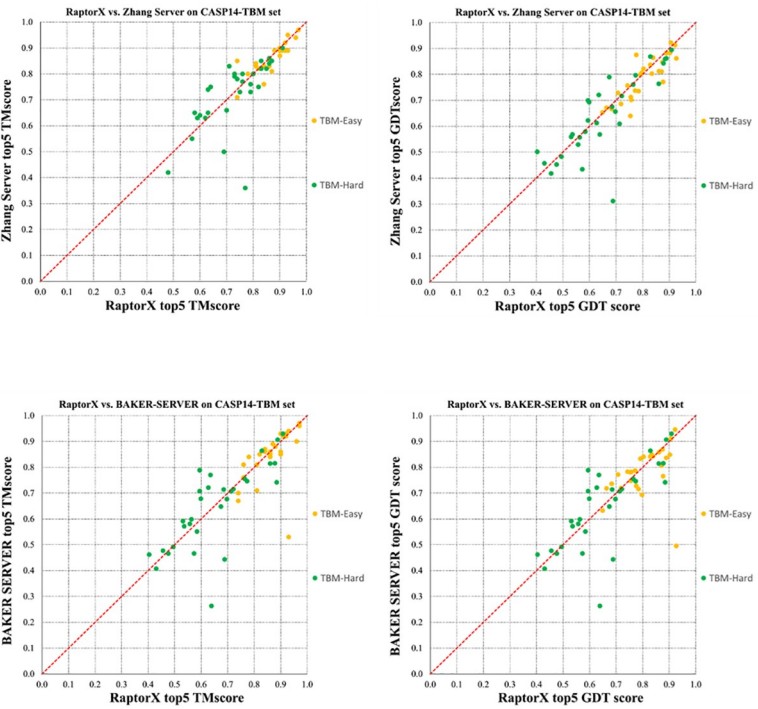

**Fig 6. The head-to-head comparison between RaptorX, Zhang-Server, BAKER-RosettaServer on the CASP14 TBM targets.** The best of top 5 models are evaluated. Top left: RaptorX vs. Zhang-Server in terms of TMscore. Top right: RaptorX vs. Zhang-Server by GDT. Bottom left: RaptorX vs. RosettaServer by TMscore. Bottom right: RaptorX vs. RosettaServer by GDT. Each point represents two 3D models generated by two competing methods. GDT is scaled to [0, 1].

did particularly well on two targets T1030-D1 and T1099-D1. The results in Table 5 and Fig 6 are summarized from the CASP14 official results. The t-test shows that neither the difference between RaptorX and Zhang-Server is statistically significant (P-value = 0.736 when the best of top five models are considered) nor is the difference between RaptorX and RosettaServer (P-value = 0.597), possibly because the number of test proteins is small.

By the way, both Zhang-Server and Baker-RosettaServer and the top human group Alpha-Fold2 have implemented a similar idea in CASP14, i.e., feeding templates into deep neural networks to help model a TBM target. In addition to using templates, AlphaFold2 did much better by directly predicting atom coordinates instead of inter-atom distance distribution, employing a Transformer-like deep neural network and possibly other techniques. Note that for some TBM targets (especially TBM-hard targets), without using templates our template-free modeling method may generate 3D models of similar or higher quality than using templates. For example, for T1047s2-D2, T1065s1-D1, T1083-D1 and T1084-D1 and T1085-D3, our template-free modeling method predicted 3D models with GDT 88.86, 88.44, 87.77, 90.84 and 82.89, respectively. Since it is unclear which specific techniques are used by other groups on a specific target, here we calculate the performance of the three servers on all the TBM targets.

## Discussion

We have presented two new methods DRNF and NDThreader for TBM. DRNF uses a deep convolutional residual neural network (ResNet) and CRF (Conditional Random Fields) to

predict sequence-template alignment from sequential features and NDThreader uses predicted distance potential to further improve the alignments generated by DRNF. Our test results on the CASP13 and CAMEO data show that our methods can generate much better alignment and have better threading performance than existing methods, especially when very similar templates are not available. When predicted distance information is not used, DRNF can generate much better alignments than those methods that mainly rely on sequence profiles such as HHpred and CNFpred. When predicted distance potential is used NDThreader outperforms those methods that use predicted contact and/or distance such as CEthreader and DeepThreader. The methods presented here can also be used to align two proteins without solved structures, as long as we replace the native structure information of a template with predicted structure information. Instead of using MODELLER and RosettaCM to build 3D models from an alignment, we have also presented our own method to build 3D models from both alignments and sequence coevolution information. The blind test in CASP14 confirmed that our method for protein alignment and 3D model building works well for TBM targets. One potential issue with our method is that several key modules are implemented independently. It may further improve modeling accuracy if we can build an end-to-end system that takes a pair of query protein and template as input and directly outputs the 3D model of the query. The ResNet method used in this work can also be replaced by GNN (Graphical neural network) to make use of the template structure in a better way.

## Methods

### Training and test data

**Training and validation data.** We constructed the training and validation data based upon a PDB40 database dated in August 2018, in which any two protein chains share <40% sequence identity. See S4 File for detailed protein list. Below is the procedure.

1. Assign superfamily IDs to all protein chains in PDB40 based upon their classification in SCOP version 2.06 [25] created in February 2016. One multi-domain protein chain may have multiple superfamily IDs. A protein chain is discarded if it is not included in this SCOP, which implies that all our training proteins were deposited to PDB before 2016.

2. Divide all the protein chains into groups by their superfamily IDs so that proteins in one group share one common superfamily ID. A multi-domain protein chain may belong to multiple groups. If a group has more than 200 proteins, we just keep 200 proteins by random sampling. If a group has fewer than 20 proteins, we merge it with another group sharing the same fold ID. In total we obtain about 400 groups and all proteins in one group are either in the same superfamily or have a similar fold.

3. Run DeepAlign [26] to calculate the structure similarity of any two proteins in the same group. Keep only the protein pairs with structure similarity (i.e., TMscore) between 0.45 and 0.95.

4. Divide all protein pairs into 4 groups by their structure similarity: (0.45, 0.65], (0.65, 0.75], (0.75, 0.85] and (0.85, 0.95]. Randomly sample protein pairs in each group so that the number of protein pairs in these groups are approximately in the ratio 1:2:2:1. That is, we emphasize more on those protein pairs at medium similarity level since for very similar proteins existing tools HHpred and CNFpred are good enough and for dissimilar proteins template-free modeling may work better.

5. Finally, we obtain ~190000 protein pairs, from which we randomly select 8000 for validation and 1000 to form our in-house test set. The test protein pairs are selected so that they

are not similar to any training/validation protein pairs. We say two protein pairs are similar if their query proteins share at least one superfamily IDs and so do their template proteins.

## Test data for evaluating alignment accuracy

1. An in-house test set consists of 1000 protein pairs as described above. The proteins in this set have length from 32 to 655 and their structure similarity (TMscore) ranges from 0.45 to 0.95. This set is mainly used to test DRNF.

2. CASP13 data. We use 112 official-defined domains with publicly available experimental structures. We run TMalign to find their structurally similar proteins (with TMscore>0.5) in the PDB40 database. We select up to top 5 similar templates for the FM and FM/TBM targets and top 10 similar templates for the TBM-Hard and TBM-Easy targets to form 765 protein pairs. Many protein pairs in this set have proteins of very different length, which makes it very challenging to build accurate sequence-template alignments.

3. CAMEO set. We select 131 CAMEO targets released after 2018 which are not similar to our training/validation proteins. We Run TMalign to find their similar templates (TMscore>0.5) in PDB40 excluding those templates sharing >40% sequence identity. Afterwards, for each CAMEO target, we select up to 10 most similar templates to form ~840 sequence-template pairs.

Finally, we use two structure alignment tools TMalign and DeepAlign to generate reference alignments for a protein pair.

**Test data for threading.** We test the threading performance using the CASP13 set. It consists of 112 officially defined domains. These domains are divided into 4 categories by their difficulty level: FM (template-free modeling targets), FM/TBM, TBM-hard (hard template-based modeling targets) and TBM-easy (easy TBM targets) [27]. We use the PDB40 dated in May 2018 as our template database. Any two proteins in PDB40 share <40% sequence identity. We also examined the performance of our method in CASP14, in which PDB70 was used as the template database.

## Evaluation method

**Evaluate alignment accuracy.** We calculate both reference-dependent and reference-independent alignment accuracy. To calculate reference-dependent accuracy, we use the structure alignment of two proteins as our reference alignment since usually structure alignment is more accurate than alignments generated by a threading method. There are many structure alignment tools and here we use TMalign and DeepAlign to build two different reference alignments for a protein pair. We use recall and precision to evaluate the reference-dependent accuracy. Precision is defined as the percentage of correctly aligned positions judged by the reference alignments. Recall is the percentage of aligned positions in the reference alignment that are also aligned by a threading method. For reference-independent evaluation, we build a 3D model for the query protein using MODELLER [21] based on its sequence-template alignment generated by a threading method and then evaluate the quality of the 3D model mainly by TMscore and GDT. TMscore ranges from 0 to 1 and GDT ranges from 0 to 100. The higher the score, the better the model quality.

**Evaluate threading performance.** We evaluate threading performance by measuring the quality of 3D models built by MODELLER from the first-ranked, the best of top 5 templates and for hard targets the best of top 50 templates. This allows us to study how well we may select the best templates for hard targets.

## Protein features

We use the following sequential features to predict alignment score between a query residue and a template residue.

1. *amino acid identity*. It is 1 if the two residues are the same, otherwise 0.

2. *amino acid substitution matrix*. To handle proteins at different similarity levels, we use three amino acid substitution matrices BLOSUM80, BLOSUM62 and BLOSUM45 [28] to score the similarity of two residues.

3. *sequence profile similarity*. We calculate this by the inner product of PSFM (position-specific frequency matrix) and PSSM (position-specific scoring matrix) in two directions: query PSFM to template PSSM and template PSFM to query PSSM. Both PSFM and PSSM of all test proteins are derived from the profile HHM built by HHblits [29] with E-value = 0.001 and uniclust30 dated in October 2017.

4. *Secondary structure score*. We predict the 3-class and 8-class secondary structure types of the query protein using RaptorX-Property [30] from its sequence profile, calculate the template secondary structure using DSSP [31], and then calculate secondary structure similarity between the query and template proteins.

5. *Solvent accessibility*. We use RaptorX-Property to predict the solvent accessibility of the query protein from its sequence profile, and DSSP [31] to calculate the solvent accessibility of the template.

We also use predicted distance information of the query protein and the native distance matrix of the template. We use the deep ResNet method described in [6] to predict $C_\beta$-$C_\beta$ discrete distance distribution for a query protein sequence and convert it to distance potential using the DFIRE reference state [32]. We discretize inter-atom distance into 14 intervals: < 4Å, 5 to 6Å,. . ., 14 to 15Å, 15-16Å and >16Å. Distance potential is used to quantify how well a pair of sequence residues can be aligned to a pair of template residues. While predicting distance potential, multiple sequence alignments (MSAs) are built from a sequence database created before March 2018 to ensure a fair comparison with other methods on the CASP13 and CAMEO data.

## Representation of protein alignment

Let $T$ denote a template protein with a solved structure and $S$ a query protein sequence under prediction. Let $A = \{a_1, a_2, a_3, . . ., a_L\}$ denote an alignment between $T$ and $S$ where $L$ is the alignment length and $a_i$ is one of the five states $M, I_x, I_y, G_h$ and $G_t$. $M$ represents two residues being aligned, $I_x$ and $I_y$ represent insertion at the template and the query proteins, respectively. $G_h$ and $G_t$ represent the head and tail gaps, respectively. As shown in Fig A in S1 File, one alignment can be represented as a sequence of $L$ states, a path in the alignment matrix, and a sequence of $L$ triples. Each triple consists of two residue indices and one state. Here residue index ranges from 0 to sequence length minus 1, and -1 is used to indicate head and tail gaps. An alignment can also be represented by a set of $5N_1N_2$ binary variables:

$$\{z_{ij}^u : -1 \leq i \leq N_1 - 1, -1 \leq j \leq N_2 - 1, u \in \{M, I_x, I_y, G_h, G_t\}\}$$

Where $N_1$ and $N_2$ are protein lengths, and $z_{ij}^u$ is equal to 1 if and only if the alignment path passes (i, j) with state $u$, i.e., the triple (i, j, u) appears in the triple representation of the alignment.

## DRNF for protein alignment without distance information

Our DRNF method uses two 1D deep ResNet, one 2D deep ResNet and one CRF (Conditional Random Fields), as shown in Fig B in S1 File. The 1D ResNet extracts sequential context of one residue in the template and query proteins and the 2D ResNet extracts pairwise context of a residue pair (one query residue and one template residue) and predicts the alignment score of this pair of residues. Outer concatenation is used to convert 1D sequential information to 2D pairwise information. The 1D ResNet consists of 10 convolutional layers and the same number of instance normalization layers and RELU layers. The kernel size of a 1D convolutional layer is 3. The 2D ResNet is more important, consisting of 20 residual blocks, each having 3 convolutional layers, 3 instance normalization layers and 3 ReLU layers. We use $5 \times 5$ as the kernel size of a 2D convolution layer. To make a better use of GPU, we group the training protein pairs into minibatches by their length product. One minibatch may contain multiple pairs of small proteins (e.g., $150 \times 150$) or only one pair of two large proteins (e.g. $600 \times 600$). We have also tested a few other slightly different network architectures such as adding 1D LSTM onto 1D ResNet, but have not observed any significant performance again.

Deep ResNet can predict the alignment score of any two residues of the query protein and the template. To produce a complete sequence-template alignment, we employ CRF, a probabilistic graphical model that takes the alignment score produced by deep ResNet as input. CRF also needs a state transition matrix to score the transition from one state (e.g., $M$) to the other (e.g., $I_x$). As shown in Table A, there are 12 feasible state transitions and 13 forbidden transitions. Since we want to generate a local alignment, we do not penalize the head and tail gaps and thus, set the score of the following state transitions to 0: match to tail gap, head gap to match, head gap to head gap, tail gap to tail gap. In addition, we do not allow a direct transition from a head gap to a tail gap, which implies that each alignment shall contain at least one pair of aligned residues. To avoid generating multiple equivalent alignments, we allow only $I_x \rightarrow I_y$ but not $I_y \rightarrow I_x$. That is, when both insertions and deletions appear in the same region, we always place $I_x$ before $I_y$.

We may use two methods to build an alignment based upon the CRF model: Viterbi [33] and MaxAcc (maximum expected accuracy) [34]. Viterbi generates the alignment with the highest probability while MaxAcc produces the alignment with the maximum expected accuracy. Both methods have time complexity proportional to the product of two protein lengths, but MaxAcc takes approximately twice the running time of Viterbi.

We train DRNF by maximum-likelihood, i.e., maximizing the probability of the reference alignments of the training protein pairs. We find out that even if fixing the state transition matrix, we can still obtain a very good DRNF model.

## Protein alignment with predicted distance potential

When templates are not very similar to a target, we use DRNF to generate initial alignments (without using distance information) and then employ predicted distance potential to improve them. With predicted distance potential, we may score a sequence-template alignment $A$ as follows.

$$S = w \times S_{singleton} + S_{pairwise} = w \times \Sigma \neg_{(i,j,u) \in Z} \theta_{ij}^u z_{ij}^u + \Sigma \neg_{(i,j,u) \in Z, (k,l,v) \in Z} \theta_{ij}^u z_{ij}^u z_{kl}^v \ s.t. \Sigma \neg_{j,u} z_{ij}^u$$

$$= 1 \ for \ any \ i \tag{1}$$

Where w is a weight factor with a default value 1. When the template is very similar to the query (which can be roughly determined by HHblits E-value $<$ 1E-10), we may use a larger value for w (e.g., 20). $z_{ij}^u$ is a binary variable that equals to 1 if and only if the triple (i, j, u) is in

the alignment $A$ (see the representation of an alignment). $\theta_{ij}{}^{u}$ represents the score generated by DRNF for residues i and j with state u. $\theta_{ijkl}{}^{uv}$ is equal to 0 if either $u$ or $v$ is not the match state. Otherwise, it equals the potential of query residues j and l falling into a distance bin $d$ where $d$ is the distance bin into which the two template residues i and k fall. As mentioned before, the distance potential is predicted by our deep ResNet method described in [6].

To find an alignment maximizing Eq (1) is computationally hard. We have developed two methods to improve alignments using predicted distance potential: ADMM (Alternating Direction Method of Multipliers) and deep ResNet. They can be iteratively and alternately used to improve alignments. The detailed ADMM method is described in [13,19] and section E in S1 File. Briefly speaking, ADMM starts from an initial alignment and iteratively improves it by incorporating distance potential. ADMM usually converges to a local optimum and thus, may not be able to find the best alignment. To overcome this, we initialize ADMM by 4 different initial alignments generated by four different DRNF models trained with two different input features (whether predicted secondary structure and solvent accessibility are used or not) and two different reference alignments (generated by TMalign or DeepAlign). Our experimental results show that with 4 different initial alignments on average we may improve the alignment quality by 0.01~0.02 TMscore over a single initial alignment.

Besides ADMM, we have trained one deep ResNet to improve an alignment with predicted distance potential, using the same training set as DRNF. The input of this ResNet includes an initial alignment generated by DRNF or ADMM, the alignment score $\theta_{ij}{}^{u}$ (see Eq (1)) generated by DRNF for any two residues i (in template) and j (in query), and the distance potential score between i and j. The initial alignment is represented as a binary matrix of dimension *template length × target length*. Given an initial alignment, we may calculate the distance potential score between i and j by summing up $\theta_{ijkl}{}^{uv}$ (see Eq (1)) over all (k, l) where template residue k is aligned to query residue l in the initial alignment and the Euclidean distance between i and k is less than 16Å. Each input information can be represented as a matrix of shape *template length × target length*, which is why we may use a 2D ResNet (similar to what is used in DRNF) to predict a query-template alignment from the input. In summary, this deep ResNet method differs from ADMM in that the latter uses a linear combination of $\theta_{ij}{}^{u}$ and $\theta_{ijkl}{}^{uv}$ (i.e., Eq (1)) while the former uses a neural network to integrate $\theta_{ij}{}^{u}$ and $\theta_{ijkl}{}^{uv}$.

It is possible to use machine learning to select the best alignment for a protein pair. Here we use the following score.

$$Selection\ Score = w_{\neg_1} \times S_{singleton} + S_{pairwise} + w_2 \times S_{norm}$$

Where $S_{singleton}$ and $S_{pairwise}$ are defined in Eq (1) and $S_{norm}$ is $S_{pairwise}$ normalized by the number of aligned positions; $w_1$ is set to 1 by default and can be elevated to 20 for easy targets; and $w_2$ has a default value 5. By using the normalized distance potential, we emphasize more on the quality of an alignment instead of the alignment length and thus, avoid generating a lengthy alignment for two large proteins in which many aligned positions are of low quality.

## 3D model building from alignments

In our self-benchmarking, we build 3D models from sequence-template alignments using MODELLER since it runs very fast and can finish a large-scale test very quickly. Nevertheless, in the CASP14 blind test, except when a test target shares >40% sequence identity with its templates (where MODELLER was used), we built 3D models from a sequence-template alignment using our own folding engine originally developed for template-free modeling [20]. To fulfill this, we fed a sequence-template alignment, the template distance matrix and the sequence coevolution into a 2D ResNet (of 100 2D convolutional layers) to predict inter-

residue orientation/distance distribution and then converted this distribution into distance/orientation potential, which is then fed into PyRosetta to build 3D models by minimizing the potential. The idea of feeding templates into a deep neural network was initiated by our group in CASP13 [5] and now has been adopted by quite a few groups in CASP14 such as Alpha-Fold2 and Rosetta. Different from AlphaFold2 and Rosetta that used multiple templates to build one 3D model, in CASP14 we used only one template to build one 3D model since our multi-template modeling has yet to be implemented.

## Supporting information

**S1 File. The supplementary file for more method details and auxiliary results.**
(DOCX)

**S2 File. The detailed threading results on the CASP13 targets and the results of some top CASP13 servers.**
(XLSX)

**S3 File. The detailed results of three top servers on the CASP14 TBM targets.**
(XLSX)

**S4 File. A zip file containing the lists of training, validation, and test protein pairs.**
(ZIP)

## Author Contributions

**Conceptualization:** Jinbo Xu.

**Data curation:** Fandi Wu, Jinbo Xu.

**Formal analysis:** Fandi Wu, Jinbo Xu.

**Funding acquisition:** Jinbo Xu.

**Investigation:** Fandi Wu.

**Methodology:** Jinbo Xu.

**Project administration:** Jinbo Xu.

**Resources:** Jinbo Xu.

**Software:** Fandi Wu, Jinbo Xu.

**Supervision:** Jinbo Xu.

**Validation:** Fandi Wu.

**Visualization:** Fandi Wu.

**Writing – original draft:** Fandi Wu, Jinbo Xu.

**Writing – review & editing:** Fandi Wu, Jinbo Xu.

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
