## [Decision Letter · Decision Letter 0]

15 Feb 2021

Dear Dr. Xu,

Thank you very much for submitting your manuscript "Deep Template-based Protein Structure Prediction" for consideration at PLOS Computational Biology.

As with all papers reviewed by the journal, your manuscript was reviewed by members of the editorial board and by several independent reviewers. In light of the reviews (below this email), we would like to invite the resubmission of a significantly-revised version that takes into account the reviewers' comments.

Both reviewers clearly appreciate the reported performance and the clarity of the manuscript. However, the second reviewer raises an important concern about the question of reproducibility and method availability. Currently only a link towards the RaptorX server is provided, but all seems to be integrated and not independently testable. Please address this point carefully.

We cannot make any decision about publication until we have seen the revised manuscript and your response to the reviewers' comments. Your revised manuscript is also likely to be sent to the same reviewers for further evaluation.

Sincerely,

Martin Weigt

Guest Editor

PLOS Computational Biology

Arne Elofsson

Deputy Editor

PLOS Computational Biology

Both reviewers clearly appreciate the reported performance and the clarity of the manuscript. However, the second reviewer raises an important concern about the question of reproducibility and method availability. Currently only a link towards the RaptorX server is provided, but all seems to be integrated and not independently testable. Please address this point carefully.

Reviewer's Responses to Questions

**Comments to the Authors:**

Reviewer #1: The authors improve template-based modeling over their previous method by using ADMM to align predicted distance potential to that of a template. Overall good description of method and what was done during CASP.

Some general concerns:

1) The authors state that "DeepThreader is the first distance-based threading". But it seems DALI (Distance-matrix ALIgnment method) could also be used to align distance matrices. Have authors tried using predicted distance matrices with DALI for template search/ranking and threading?

2) For cases where DeepThreader did better than NDThreader, is this because NDThreader got stuck in local minimum during ADMM alignment? If the best possible alignment (using TMalign) or alignment from DeepThreader is rescored using NDThreader's score function, does it have a better or worse score than the NDThreader's solution?

3) For Table 5, it would be good to see a scatterplot comparing the methods. Averages can be a bit misleading, as a single outlier target can skew the results. It would be good to know what fraction of the targets RaptorX did better than Zhang-Server/BAKER-SERVER. The statement in the abstract "the best GDT score among allCASP14 servers on the 58 TBM targets." makes it sound like the method got the best GDT score on all the 58 targets... though I'm guessing this only true on average?

Reviewer #2: The authors of "Deep Template-based Protein Structure Prediction" present a highly successful deep learning method for improving the accuracy of template based modelling of protein structures.

The paper is written in a readable, methodical way and is sufficiently easy to follow for an interested reader. The methods presented are both interesting and sufficiently novel.

They would be of great use to the general community, if only it were available. In a current format, the article describes the method, though in a way that is not sufficient to replicate it by the reader.

= Major concerns:

1. The paper introduces new architecture - Deep Convolutional Residual Neural Fields (DRNF), which appears exciting and (judging by authors' report) performs exquisitely well for the problem approached. However, there is no reference implementation of this architecture available, not to speak about the actual method that could be used on arbitrary data. Without ability to replicate results, the message of the paper becomes much weaker.

2. Authors provide the compositions of the training/validation/testing set, but these are all the data provided. As there is no method available, there is no way to verify the correctness of the results, nor there is a way to test the method independently. Furthermore, there is no way to assess the quality of predictions beyond the metrics provided by the authors.

3. Authors of AlphaFold2 claim that their method largely supersedes TBM methods for protein structure prediction. How does DRNF and NDThreader compare to such a method - especially in terms of practical usefulness for the protein structure prediction and modelling communities?

4. Training/validation/test set separation. In p. 16 l. 273. "A multi-domain protein chain may belong to multiple groups". It is not evident for me, that such a partition does not result in an information bleed between data sets, especially in context of page 17 l. 285-288. Let's assume one protein pair contains domains in superfamilies S1 and S2, and is assigned to (let's say) training set by virtue of domain S1. Let's assume also that superfamily S2 is not a part of the training set (which, in light of the paper it does not need to be). Can the same protein pair be assigned to the validation/test set by virtue of domain S2? Can proteins containing domains in S2 be a part of validation/test sets, even though examples of these domains are already in training set?

5. ADMM diversification, cf. p. 22. l. 411-413. How much of divergence can one expect with different initialization criteria? It would seem, like for the "easier" tasks inclusion of SAS and/or SS priors should be irrelevant. And both TMalign and DeepAlign should result in identical alignments... Can you provide an estimate how often do such a treatment help?

= Minor concerns:

p. 1 l. 12/13 NDThreader becomes DNThreader

p. 1. l. 17 unnecessary space in 'co-evolution'

p. 2. l 30 "methods developed for TBM": missing verb

p. 2. l. 40 "good percentage" - would you care to specify?

p. 15. Table 3. for NDThreader at (TM+GDT)/2 "top 1" score value 0.419 is incongruent with the rest of the table. I assume it is a typo... Additionally, please look into the table headers and their alignment.

p. 22. l. 410. "converges to local optimal" - I think you mean "optimum"

p. 22. l. 411-413. How much of divergence can one expect with different initialization criteria? It would seem, like for the "easier" tasks inclusion of SAS and/or SS priors should be irrelevant. And both TMalign and DeepAlign should result in identical alignments...

**Have all data underlying the figures and results presented in the manuscript been provided?**

Reviewer #1: Yes

Reviewer #2: **No: **Authors did not provide the results they discuss, nor did they provide the way to replicate them. The methods used in the paper remain private and as such none of the results can be verified or replicated.

PLOS authors have the option to publish the peer review history of their article (what does this mean?). If published, this will include your full peer review and any attached files.

Reviewer #1: No

Reviewer #2: No
---

## [Decision Letter · Decision Letter 1]

30 Mar 2021

Dear Dr. Xu,

Thank you very much for submitting your manuscript "Deep Template-based Protein Structure Prediction" for consideration at PLOS Computational Biology. As with all papers reviewed by the journal, your manuscript was reviewed by members of the editorial board and by several independent reviewers. The reviewers appreciated the attention to an important topic. Based on the reviews, we are likely to accept this manuscript for publication, providing that you modify the manuscript according to the review recommendations.

Besides a number of very minor comments, Reviewer 2 raises major concerns with respect to the sharing of your code, which does not comply with open source standards. Please change access to the software such that it can be downloaded and used without registration. For your information, the PLoS standards for software sharing are reproduced here:

"

We expect that all researchers submitting to PLOS submissions in which software is the central part of the manuscript will make all relevant software available without restrictions upon publication of the work. Authors must ensure that software remains usable over time regardless of versions or upgrades. If the original software is not able to be shared, authors must provide a reasonable facsimile.

[https://journals.plos.org/ploscompbiol/s/materials-and-software-sharing#loc-sharing-software].

If the Software is a central part of the submission the paper must meet the following requirements:

Based on open source standards

Conform to the Open Source Definition

Deposited in an open software archive (see “Depositing software,” below)

Included in the submission as supporting information

Linked directly from the manuscript file

"

Sincerely,

Martin Weigt

Guest Editor

PLOS Computational Biology

Arne Elofsson

Deputy Editor

PLOS Computational Biology

[LINK]

"We expect that all researchers submitting to PLOS submissions in which software is the central part of the manuscript will make all relevant software available without restrictions upon publication of the work. Authors must ensure that software remains usable over time regardless of versions or upgrades. If the original software is not able to be shared, authors must provide a reasonable facsimile."

[https://journals.plos.org/ploscompbiol/s/materials-and-software-sharing#loc-sharing-software].

If the Software is a central part of the submission the paper must meet the following requirements:

Based on open source standards

Conform to the Open Source Definition

Deposited in an open software archive (see “Depositing software,” below)

Included in the submission as supporting information

Linked directly from the manuscript file

Reviewer's Responses to Questions

**Comments to the Authors:**

Reviewer #1: The authors have addressed most of my concerns. I would recommend acceptance.

DALI should be cited, in the context of "first distance-based threading method", regardless if one uses real vs. predicted distances.

Reviewer #2: Thank you for comprehensive addressing of my questions. I sustain my opinion, that the work presented in this paper is of high quality, both in terms of novelty and implementation and constitutes a valuable addition to the knowledge in the field.

However, even though the Authors do provide the software presented in the paper, it does not fulfil the Open Source requirement of the Journal. It is impossible to download the software without registering an account (which compromises the implied anonymity of the review, or requires submitting fictitious personal data). Moreover, users with commercial email addresses cannot register - which is a minute detail, but still contravenes the Open Access/Open Source policy of the Journal. The software, even though it is (commendably) provided as a source code, comes with blanket reservation:

"Unless explicitly permitted by the RaptorX team, these standalone programs and datasets cannot be used for commercial purposes or included into a web server." in README file on the web server, where one can download the software. This alone makes the software not OSI-license compatible (https://opensource.org/licenses/alphabetical) which is a prerequisite for software papers in PLOS journals. The restriction of access, redistribution and usage violate the OSI Open Source Definition (https://opensource.org/docs/osd) at least in points 1, 3, 5, and 6.

Were authors to release the software with a free license, I would have no reservations.

My remaining concerns are largely cosmetic:

p. 2 "protein data bank" -> "Protein Data Bank"

p. 4 "New Deep-learning Therader" -> "... Threader"

p. 9 The captions of Table 3 are still improperly typeset - which I trust will be fixed in the editorial office, but still makes the paper less approachable to read for reviewers.

p. 15 "Run TMalign to find ... " - it feels like there is a subject missing, consider adding "We".

And general concern:

In many places authors contrast methods, asserting a supremacy of one over the other. Due to limited size of comparison sets, and often small margins of difference, it would greatly improve the paper, if there were confidence intervals or - even better - p-values provided with such comparisons.

**Have all data underlying the figures and results presented in the manuscript been provided?**

Reviewer #1: Yes

Reviewer #2: Yes

PLOS authors have the option to publish the peer review history of their article (what does this mean?). If published, this will include your full peer review and any attached files.

Reviewer #1: No

Reviewer #2: No

Figure Files:

Data Requirements:

Reproducibility:

References:

---

## [Editor Report · Decision Letter 2]

11 Apr 2021

Dear Dr. Xu,

We are pleased to inform you that your manuscript 'Deep Template-based Protein Structure Prediction' has been provisionally accepted for publication in PLOS Computational Biology.

Best regards,

Martin Weigt

Guest Editor

PLOS Computational Biology

Arne Elofsson

Deputy Editor

PLOS Computational Biology

---

## [Editor Report · Acceptance letter]

27 Apr 2021

PCOMPBIOL-D-21-00008R2 

Deep Template-based Protein Structure Prediction

Dear Dr Xu,

I am pleased to inform you that your manuscript has been formally accepted for publication in PLOS Computational Biology. Your manuscript is now with our production department and you will be notified of the publication date in due course.

With kind regards,

Katalin Szabo
